# Improving Organizational Health Literacy Responsiveness in Cardiac Rehabilitation Using A Co-Design Methodology: Results from The Heart Skills Study

**DOI:** 10.3390/ijerph17031015

**Published:** 2020-02-05

**Authors:** Anna Aaby, Camilla Bakkær Simonsen, Knud Ryom, Helle Terkildsen Maindal

**Affiliations:** 1Department of Public Health, Aarhus University, Bartholins Allé 2, 8000 Aarhus C, Denmark; casi@ph.au.dk (C.B.S.); knudryom@ph.au.dk (K.R.); htm@ph.au.dk (H.T.M.); 2Steno Diabetes Center Copenhagen, Niels Steensens vej 2, 2820 Gentofte, Denmark

**Keywords:** health literacy, organizational health literacy, cardiac rehabilitation, intervention development, co-design, needs assessment, equity in health

## Abstract

For health services, improving organizational health literacy responsiveness is a promising approach to enhance health and counter health inequity. A number of frameworks and tools are available to help organizations boost their health literacy responsiveness. These include the Ophelia (OPtimising HEalth LIteracy and Access) approach centered on local needs assessments, co-design methodologies, and pragmatic intervention testing. Within a municipal cardiac rehabilitation (CR) setting, the Heart Skills Study aimed to: (1) Develop and test an organizational health literacy intervention using an extended version of the Ophelia approach, and (2) evaluate the organizational impact of the application of the Ophelia approach. We found the approach successful in producing feasible organizational quality improvement interventions that responded to local health literacy needs such as enhanced social support and individualized care. Furthermore, applying the Ophelia approach had a substantial organizational impact. The co-design process in the unit helped develop and integrate a new and holistic understanding of CR user needs and vulnerabilities based on health literacy. It also generated motivation and ownership among CR users, staff, and leaders, paving the way for sustainable future implementation. The findings can be used to inform the development and evaluation of sustainable co-designed health literacy initiatives in other settings.

## 1. Introduction

Responding to population health literacy needs is a promising approach to counter inequity in health and healthcare delivery [1]. Health literacy is “the combination of personal competencies and situational resources needed for people to access, understand, appraise and use information and services to make decisions about health. It includes the capacity to communicate, assert and act upon these decisions.” [2]. 

The application of health literacy capabilities cannot be separated from the demand and complexity of the context in which they are used, e.g., the health system [3]. Organizational health literacy responsiveness refers to “the way in which services, organizations and systems make health information and resources available and accessible to people according to health literacy strengths and limitations” [2]. Frameworks and tools to develop or evaluate organizational health literacy responsiveness have emerged over the past decades [4], paving the way for the integration of the concept into intervention development methodologies. Farmanova et al. (2018) provide a summary of the different tools available to tackle health literacy barriers and to facilitate the promotion of health literacy at the organizational level [4]. Despite recent progress, there is little robust evidence of the effectiveness of initiatives to improve organizational health literacy responsiveness [5]. A rapid realist review by Willis et al. (2014) suggests that actions across governmental, organizational, and partnership level can boost organizational capacity to address health literacy [6]. These include strategies that build organizational commitment, create ownership and involvement, promote a culture of ongoing organizational experimentation and learning, build community support, and strengthen teams through shared responsibilities [6]. 

Featuring all of these strategies, the Ophelia (OPtimising HEalth LIteracy and Access) approach is a methodology available to help improve individual and organizational health literacy responsiveness. It was inspired by a set of well-established intervention development methodologies [7]. Intervention Mapping (IM) is a stepwise process originally designed to identify, develop, implement, and evaluate health education programs [8]. The Ophelia approach has a similar structure, but seeks, in particular, local fit by combining the individual needs assessment with local knowledge on organization and context, and by recommending continuous low-scale quality improvement cycles [9] before large-scale implementation. It also prescribes an intervention development inspired by realist methodologies’ focusing on contexts, mechanisms, and outcomes [10]. The Ophelia approach has been tested in a number of settings and is an effective and flexible guide to help identify health literacy challenges and develop and implement locally appropriate solutions [11,12,13]. However, to the best of our knowledge, the Ophelia approach has not previously been applied in cardiac rehabilitation (CR). 

People with cardiac conditions are subject to high demands on their self-care abilities, including their health literacy capabilities. They undergo complex treatment regimens that usually require extensive health behavior modifications [14]. A large Danish study has shown that people with cardiac conditions have significantly lower health literacy than the general population [15]. Among people with cardiac conditions, low health literacy is associated with adverse health behaviors [16,17] and poor quality of life [17,18,19]. CR programs are designed to sustain or improve self-care, health behavior, and quality of life, but participation and adherence is dependent on social health determinants such as education, cohabitation, and income [20]. Health literacy is associated with all of these determinants [21] and may be a modifying factor in relation to their impact on health outcomes [22,23]. Organizational initiatives responding to health literacy needs may be a suitable approach to improve the equitable impact of CR services [24,25].

With the ultimate goal of improving organizational health literacy responsiveness in a municipal CR setting, the Heart Skills Study aimed to: (1) develop and test an organizational health literacy intervention using an extended version of the Ophelia approach, and (2) evaluate the organizational impact of the application of the Ophelia approach.

## 2. Materials and Methods 

### 2.1. The Ophelia Approach

The Ophelia approach is a systematic intervention development and testing methodology and entails a series of steps (Figure 1) [7,26]. To allow flexibility and creativity in local settings while maintaining the central values of the Ophelia approach, a set of core principles guides its application (Table 1) [11]. 

In the following section, we present how the seven steps of the Ophelia approach were applied in the Heart Skills Study and subsequently how we carried out our examination of organizational impact of the process.

### 2.2. Application of the Ophelia Approach

#### 2.2.1. Aim, Scope, and Setup (Step 1) 

The Heart Skills Study was carried out between January 2017 and November 2019. It was set in Randers Municipal Rehabilitation Unit in Randers Municipality, Denmark, an area with approximately 98,000 inhabitants. The Unit is situated in a community health center along with several other health services and clinics. The unit offers rehabilitation programs covering major chronic conditions or post-hospitalization rehabilitation as part of the Danish free-of-charge public health insurance system. The Heart Skills Study was carried out by the CR team, which consisted of a nurse, three physiotherapists, an occupational therapist, and a dietician. 

The study focus, scope, and overall aim were developed by leaders in the rehabilitation unit and the research team prior to the co-design process. A project management team, consisting of a development physiotherapist, two user representatives, and a researcher (A.A.), facilitated the Heart Skills Study co-design process. 

#### 2.2.2. Needs Assessment (Step 2)

We completed two distinct local needs assessment analyses: A user health literacy assessment and an organizational health literacy responsiveness analysis. The latter was included as an add-on to the original Ophelia approach.

For the user health literacy assessment, we carried out a cross-sectional survey among all 222 people referred to CR in Randers Municipal Rehabilitation Unit in 2017. The survey is described in detail elsewhere [27]. Adhering to the Ophelia approach [26], we based our survey on the Health Literacy Questionnaire (HLQ). The HLQ consisted of 44 items divided into nine scales each covering one of the following aspects of health literacy: (1) Feeling understood and supported by healthcare providers, (2) having sufficient information to manage health, (3) actively managing my health, (4) social support for health, (5) active appraisal of health information, (6) ability to actively engage with healthcare providers, (7) navigating the healthcare system, (8) ability to find good health information, and (9) understanding health information well enough to know what to do. The HLQ was thoroughly validated [28,29] and was translated into Danish using standardized procedures [30]. 

We analyzed the survey data using hierarchical cluster analysis. This provided further details on different health literacy profiles within the population than is available in the original survey study [27]. The cluster analysis was based on all nine HLQ scale scores using Square Euclidian Distance as the distance measure and Ward’s linkage as the clustering method [31]. Since the HLQ scales use two different response ranges (1–4 and 1–5), all scores were converted to z-scores when reported to allow direct comparison between scales. Based on the clusters, we drafted short vignettes (narratives) representing the typical health literacy profiles. Eight short semi-structured telephone interviews with representatives from the three most challenged clusters were conducted to provide examples of life conditions and health literacy challenges. This information was then anonymized, processed, and combined in the final vignettes [7,11]. The vignettes were developed before the data were converted to z-scores. Individuals interviewed were not relocated to other clusters after conversion.

We based our organizational health literacy responsiveness analysis on the Organizational Health Literacy Responsiveness Self-assessment Tool and Process (Org-HLR) [32,33], which was carried out across the rehabilitation unit (not only the CR team). The Org-HLR comprised three consecutive workshops and associated tools: (1) A two-hour reflection workshop in which staff and leaders from the rehabilitation unit familiarized themselves with the concept of health literacy and applied it to their local context, (2) a four-hour self-evaluation workshop in which staff evaluated their organization’s health literacy responsiveness and came up with improvement ideas, and (3) a three-hour prioritization workshop in which staff representatives (*N* = 4) and leaders (*N* = 3) prioritized their ideas in order to inform a future action plan to improve their organizational health literacy responsiveness. Our methodology and analysis are reported in detail elsewhere [34].

#### 2.2.3. Idea Generation, Program Logic Model, and Intervention Details (Steps 3–5)

In order to generate ideas derived from the needs assessment, we continued the process conducting three separate idea-generating workshops for staff from the cardiac team, leaders from the rehabilitation unit, and users of the CR service. Users were recruited by convenience and interest by the CR nurse among those enrolled in the CR program, while all staff and leaders available at the time participated. In each workshop, the vignettes were presented and participants were asked to consider how each of the people described could be supported at an individual and organizational level. After the workshop, we categorized each of the emerging ideas under one of the following themes: Program referral and start-up, program activities, patient education and information, relations with healthcare providers, social support, external collaborators, program completion, and any other ideas. 

As part of the co-design process, a crude prioritization of ideas for interventions was subsequently made by the project management team. Guided by the overall aim of the Heart Skills Study (step 1), we first developed specific intervention objectives. From the workshop ideas and the Org-HLR results we then choose preliminary intervention elements. Based on these and a systematic literature assessment of CR interventions, researcher A.A. developed a preliminary program logic model for two novel interventions. During the subsequent planning workshop, the program logic model was presented simultaneously to staff, leaders, and users to be discussed and allow changes and adjustments. At the same workshop, the intervention elements were refined and developed in further detail. Participants in the planning workshop included all users, staff, and leaders, who participated in one of the idea-generating workshops and were willing to continue their participation.

#### 2.2.4. Testing and Implementation (Steps 6 and 7) 

The Ophelia approach recommends the use of quality improvement cycles to test and adjust interventions. In spring 2019, each intervention element was tested in small scale in the rehabilitation unit or with relevant collaborators. We kept each test separate to be able to study the effect of the elements independently. Our results were discussed among staff, leaders, users, and collaborators at a follow-up workshop and relevant adjustments were made. Participants in the follow-up workshop included all users, staff, and leaders, who participated in one of the idea-generating workshops and were willing to continue their participation. The full and adjusted intervention was then tested in a second improvement cycle during a two-month study period in autumn 2019. 

### 2.3. Evaluating the Organizational Impact of the Intervention

We evaluated the organizational impact of the application of the Ophelia approach based on the eight Ophelia principles Table 1) [11]. This allowed us to link the evaluation directly to the aims and values underlying the Ophelia approach. 

We conducted a focus group discussion with the staff on the cardiac team that had been most involved in the Heart Skills Study (*n* = 3) as well as semi-structured individual interviews with leaders above day-to-day managerial level in the rehabilitation unit (*n* = 2) and both user representatives on the project management team (*n* = 2). In the following text the seven participating individuals are termed “participants”. To ensure validity, the interviews were conducted (C.B.S. and K.R.), transcribed verbatim (C.B.S.), and analyzed (C.B.S.) by researchers who did not facilitate the co-design process (A.A.). The interview guides were based on the eight Ophelia principles and are available in the Appendix A.

The organizational impact analysis was based on a deductive closed coding (i.e., preselected codes) [35]. The codes corresponded to the eight Ophelia principles. Each interview was coded separately, then data was merged to examine similarities and differences across participant type (staff, leaders, and users), and, finally, we synthesized and summarized the findings and highlighted major points using participant quotes.

### 2.4. Approvals and Ethical Considerations 

The Heart Skills Study adhered to the General Data Protection Regulation (GDPR), the Helsinki Declaration, and national consent guidelines. The study was approved by the Danish Data Protection Agency (2015-57-0002 (62908, 141)). 

All survey respondents in the user health literacy assessment were informed about the aims of the study and gave verbal consent before the questionnaire was distributed. Their voluntary completion and return of the survey questionnaires constituted implied consent. 

In the organizational impact analysis, written informed consent was obtained from all participants. Because of the small number of participants and their active roles in the Heart Skills Study, complete anonymity could not be provided. This was accepted by all participants.

## 3. Results

### 3.1. Application of the Ophelia Approach

Figure 2 provides an overview of the different steps we undertook to develop and test our intervention using the Ophelia approach.

#### 3.1.1. Aim, Scope, and Setup (Step 1) 

Following a few rounds of iteration, leaders in the rehabilitation unit and the research team decided that the specific aim of the intervention development should focus specifically on vulnerable groups. The aim was phrased ”to develop a specific intervention taking off from the concept of health literacy and aiming to improve the quality of CR services offered to vulnerable individuals or groups.” 

#### 3.1.2. Needs Assessment (Step 2) 

The first part of the needs assessment was the user health literacy assessment. There were 178/222 (80.2%) people referred to CR in 2017 who responded to the survey. Of these, 162 respondents provided enough information for their data to be included in the cluster analysis. Table 2 outlines the results. We chose a seven-cluster solution based on variance within clusters and diversity between clusters. One of the clusters which appeared already in very low cluster solutions was very small. Due to data protection, we merged this cluster with its closest fit (clusters 1 and 2 in Table 2).

Each cluster had its own socio-demographic composition. For example, mean age was highest in cluster 4 (71.87 years), and there was a larger likelihood of being female (46.7%), living alone (46.7%), and having ≤ 11 years of schooling (38.5%) in cluster 7 (representing lowest overall health literacy) compared to other clusters. Clusters 1 and 2 (representing highest overall health literacy) had the highest percentage of male gender (75.86%) and the lowest percentage having ≤11 years of schooling (17.2%). 

In general, there was a tendency towards more adverse health outcomes in the clusters with the most health literacy challenges. For example, cluster 7 had the highest percentage not participating in rehabilitation (33.3%), the highest percentage of smokers (46.7%), and the lowest mean physical health-related quality of life score (36.6 units). Cluster 6 had the lowest mean mental health-related quality of life score (38.9 units), though this was also low in clusters 4 (42.3 units) and 7 (43.5 units).

The second part of the needs assessment was the organizational health literacy responsiveness analysis. The results are described in detail in a separate publication [34]. The process served as a local capacity development activity, which improved the general knowledge and understanding of health literacy. As a result of the process, the rehabilitation unit developed a local action plan consisting of 11 initiatives across both strategic, managerial, and practice levels in the unit. Of these, some were integrated in the subsequent Ophelia intervention development concerning CR and others were implemented alongside it and across the unit as a whole. An example was the introduction of the “Conversational Health Literacy Assessment Tool” [37,38], to identify health literacy strength and weaknesses in start-up sessions, which was integrated in the intervention package 2. Another example, was the use of health literacy as a formal local quality indicator in the unit, which was not part of the tested intervention, but served to support the integration of a health literate thinking in the unit.

#### 3.1.3. Idea Generation (Step 3)

Based on clusters 5 through 7, three vignettes were developed. These were used to present the survey results in an easily understandable way and to inspire the participants in the three co-design workshops (users *n* = 6, staff *n* = 5, leaders *n* = 3). Collectively, the workshops generated 47 unique ideas on how to improve health literacy responsiveness of the unit. For example, participants suggested comprehensive individual needs assessments in each start-up session (theme referral and start-up) and a stronger call for support from relatives (theme social support). 

#### 3.1.4. Program Logic Model (Step 4)

The project management team defined specific intervention objectives for three phases of the CR program: (1) Referral and start-up, (2) program delivery, and (3) program termination. Guided by these, the draft program logic model was developed by researcher (A.A.) and the project management team. Due to the limited time and resources available, we chose to focus mainly on the referral and start-up phases of the CR program. Table 3 (left column) provides an overview of the intervention objectives for this part of the CR program. 

#### 3.1.5. Planning Intervention Details (Step 5)

The program logic model was adjusted and adopted at the planning workshop (users *n* = 5, staff *n* = 3, leaders *n* = 3) where further co-design resulted in the two intervention packages adopted for pilot testing (Table 4).

#### 3.1.6. Pilot Testing (Step 6)

A few materials were developed prior to intervention piloting. Staff guides for all interventions were developed by a researcher (A.A.) based on workshop input and feedback from the cardiac team. Also, for intervention package 1.1 a pamphlet was developed based on workshop input and a focus group discussion with three relatives of CR attendees. 

Results from the initial quality improvement cycle (data not shown) were presented and discussed at the follow-up workshop (users *n* = 3, staff *n* = 5, leaders *n* = 3, lay counselor *n* = 1). Adjustments were made, e.g., improved distribution of written information to relatives and changes in the timing of offering lay counselors. 

In the second quality improvement cycle, all elements of both packages were tested simultaneously. The results in relation to the intervention objectives are reported in Table 3.

#### 3.1.7. Implementation (Step 7)

The Heart Skills Study was not designed to report on long-term and full-scale implementation. However, the intention to maintain the current intervention activities and develop them further is, however, evident from the organizational impact analysis as reported below.

### 3.2. Organizational Impact of the Heart Skills Study

Overall, our application of the Ophelia approach had a substantial organizational impact affecting both organizational values, service development strategies, and day-to-day practices. In the following, we report on our results by Ophelia principle (see Table 1).

#### 3.2.1. Outcomes Focused 

All participants reported that the Heart Skills Study had already or has the potential to enhance service quality in the unit. According to user representatives and a leader, the process might not directly increase skills training or improve clinical measurements but, rather, there have been improvements in organizational quality, thus supporting participation and adherence to CR services. 

The staff perceived that some CR users profited from the new initiatives. They have also personally enhanced their understanding about the health literacy needs of their users. Now, staff perceived themselves as being more persistent when they invited users to the program and that they differentiated their services to a larger extent:
”…If we want to act and help people in a good way, we have to differentiate, which we also did before, but now it is just much, much more clear and we have less “standard-package”…Sure, I think it has increased the quality.”*(Staff)*

#### 3.2.2. Equity Driven

All participants acknowledged that the identification of vulnerability based on health literacy has been a core improvement resulting from the Heart Skills Study. A leader put it thus:
“Well, you can say, that it is almost the DNA in this project…”*(Leader).*

According to the leaders, working with health literacy has increased recognition of vulnerable users and their diverse abilities to profit from health services. However, both a leader and a user representative emphasize that health literate CR users may also have unforeseen challenges that need to be met. 

The staff commented on the lack of vulnerable users involved in the co-design process, e.g., the workshops. They reflected that this type of user does not often have the resources or energy to participate. This may have affected the initiatives that were developed:
”Because I think, that we would have attained something completely different, if it was this type of users, we had involved from the beginning, right?…I just think it had been something different. But we just can’t really do that.”*(Staff)*

#### 3.2.3. Co-Design Approach

All participants were very positive about the participatory methodologies of the Heart Skills Study. They valued the inclusion of many different perspectives, through which they have become wiser and developed a greater understanding of each other. 

The leaders particularly appreciated the involvement of users, who they feel had given weight to the decision-making. They described their own role as mainly relating to the allocation of resources. However, the staff really appreciated the managerial backing, which meant they could invest time and resources in the study. 

The staff felt that they had the opportunity to influence the design of the developed initiatives while also being receptive to the other partners involved. The co-design process has become part of their day-to-day practice:
”You see, it does not just go on in the organization where you develop a service, but co-creation also happens between the user and me. You help him move on or you find out how you can move on together, but I am also supported to understand…the next patient better in the future.”*(Staff)*

To the staff and the user representatives, in particular, the co-design approach supported a sense of community and ownership of the study. This was emphasized by the use of words such as “our project” and “the band” (i.e., the project management team). Both the user representatives and staff reflected on the importance of ownership. They described how the peripheral involvement of the hospital was not enough to create engagement. As a result, the hospital did not distribute the information pamphlet (intervention package 1.1, Table 4) as consistently as intended. Thus, staff recommend that future cross-sectoral processes should involve co-creation with all stakeholders at an early stage. 

#### 3.2.4. Needs-Diagnostic Approach 

The user representatives acknowledged that the Heart Skills Study had identified some of the challenges of vulnerable users. According to one leader, the unit now works systematically with health literacy as an integrated part of practice, which has improved the staff’s ability to identify user needs. This observation was confirmed by the staff, who reflected that the concept has helped them understand users from a more holistic perspective and discover needs and vulnerabilities that may previously not have been detected:
”There are also some who surprised us, right? Where at first I would not have thought, that there might have been something. But then given the answers they gave (there was something, red.), which I would not necessarily have discovered otherwise.”*(Staff)*


#### 3.2.5. Driven by Local Wisdom 

According to both leaders and staff, the Heart Skills Study has built on local knowledge through the involvement of relevant stakeholders, including users, at different levels in the organization. Both user representatives felt that they had been able to contribute with their own experiences and that this played a substantial role in the study. 

According to the staff and leaders, the study was in line with a general user-centered approach in the municipality. Hence, it was not a new concept to work with differentiated care. However, according to one leader, the health literacy thinking adds a useful holistic and systematic approach to their work:
”… because one thing is the somatic disease itself, but another is…how it is experienced by the citizen. So you might be able to sort of build on, that you learn that those two things are connected…”*(Leader)*

#### 3.2.6. Responsiveness

Both leaders and staff confirmed that a procedure to identify users’ needs based on health literacy has been integrated into day-to-day practice. The Heart Skills Study has increased the understanding of what the organization should do to meet these needs. Health literacy could also be used to help guide financial prioritization in the unit. As one leader stated, however, if all users’ health literacy needs should be met, it would require a greater degree of differentiation than the present financial constraints allow.

The staff had several requests in relation to the development of more individualized services in the unit in the future. They stated that smaller exercise teams would be beneficial as it would allow them to respond better to users with low health literacy. Furthermore, they articulated a need for tools to manage the new user-provider interactions introduced through the Heart Skills Study. Sometimes they identified challenges which they did not have the opportunity to solve. They consequently felt incapable and unsure if they had done their job well enough:
“…I would like to have had some personal tools to conduct conversations even more professionally… how do I leave them and feel that I have done well enough…That is, to communicate and be in control of the chaotic conversation it can sometimes be.”*(Staff)*

#### 3.2.7. Systematically Applied

Beyond the new intervention packages, the Heart Skills Study has produced changes at several levels in the unit. According to a leader, health literacy and differentiated care have become key strategic priorities for the unit. This new focus has influenced the staff, who feel that differentiation has become a more legitimate topic and that the concept of health literacy has provided a professional argument for further differentiation. One leader emphasized the two aspects of the study that she would bring forward in particular:
”…the material content of the health literate way of thinking, but then I would also be interested in the methodological parts, that is the process, the design, if we could learn something from that. At any rate I am interested in how the staff, for one, become an integrated part of the process, and then how you could consider user involvement, which is already something we are discussing.”*(Leader)*

#### 3.2.8. Sustainable

All participants acknowledged the value of the Heart Skills Study and supported its continued implementation and development. The staff reported that the simultaneous organizational changes support the long-term sustainability of the initiatives. 

According to one leader, health literacy is now a theme in the professional development strategy of the municipal health administration beyond the rehabilitation unit. Health literacy awareness has been promoted to the whole health center and the staff involved in the Heart Skills Study have transferred their experience to other teams and divisions. This was confirmed by staff who experienced the Heart Skills Study as a more involving process, than other similar improvement processes:
”Because sometimes it is dictated from above what we have to do, right? But this is a project, which has spread very much up and down and sideways…”*(Staff)*

The leaders reported that health literacy is likely to be a continued focus in the unit as it is aligned with general strategies on inequity in health in the municipality. If so, municipal health policies may support the long-term sustainability of the Heart Skills Study:
“…and when I have confidence in this, it is because I think, that it connects to our politics on inequality in health, which is not just something we have been thinking about in this municipality. It is, after all, a general problem in the whole healthcare system.”*(Leader)*

Finally, the participants believed that other municipalities could benefit from the Heart Skills Study, for example, in helping health professionals and relatives achieve a greater understanding of vulnerable citizens. However, a leader emphasized that the results of the study could not be directly transferred to another context, as it would be in conflict with the study methodology, i.e., the participatory approach. 

## 4. Discussion

In this study, we reported on a well-applied co-designed intervention development process responding to local health literacy needs in a municipal CR unit. We produced feasible interventions targeting vulnerable users and facilitated their initial implementation. We also found that the intervention development process had substantial organizational impact by leading to the integration of the concept of health literacy, a familiarity with the use of participatory methodologies, and an improved focus on differentiated healthcare practices in the CR unit. 

Below we discuss central themes related to the development of health literacy interventions in general as well as our specific methodology. We also comment on the strengths and limitations of our study.

### 4.1. Systematic Development and Evaluation of Health Literacy Interventions 

In the past, most health literacy interventions targeted individual health literacy and very often only the functional level of obtaining and understanding health information [40]. This study contributes to counter the paucity of interventions targeting health literacy at the organizational level [4,5]. Health literacy interventions based on the Ophelia approach have successfully been carried out in primary care [11] and hospital settings [12]. More projects using the approach are in progress [41,42]. However, to the best of our knowledge, the Ophelia approach was not previously applied in CR, where low health literacy is a particular problem [27]. Only one other study reported briefly on the application of the eight Ophelia principles, but did not systematically evaluate the organizational impact [11].

### 4.2. Needs Assessments at User and Organizational Levels

By basing our user health literacy assessment on the nine-dimensional HLQ, we provided valuable insight into the health literacy strengths and weaknesses of a population referred to the CR services in much greater detail than most other health literacy measures would have allowed [43]. Using the cluster analysis, we were able to draw multidimensional profiles across the population providing a detailed picture of the challenges faced by specific subpopulations. The vignettes helped us share this data in an accessible way among staff, leaders, and users. Our organizational impact analysis confirmed that a new and more holistic understanding of user needs and vulnerabilities based on health literacy had been developed and integrated in the unit during the Heart Skills Study.

Our add-on to the Ophelia approach, i.e., organizational health literacy responsiveness analysis, served several purposes. Health literacy is not a commonly integrated concept in the Danish healthcare system. We, therefore, used the Org-HLR process to familiarize local providers with the concept. At the same time, the organizational inadequacies identified in the Org-HLR process served to place health literacy on the local agenda, securing managerial backing for the intervention development. According to the staff, this was crucial in allowing them to invest time and resources in the process. Both these purposes were supported by our particular choice of methodology (as opposed to other available frameworks for organizational health literacy assessment). The participatory process on which the Org-HLR is based allowed both staff and managers time to reflect on and integrate health literacy in their thinking, and the concrete output (i.e., the agreed action plan) ensured clear authorization for a sustained focus on health literacy in the future. In summary, this added feature to the Ophelia approach may be effective in building organizational commitment and promoting a culture of ongoing organizational learning, especially in settings not familiar with health literacy. Willis et al. (2014) argued that this may be a central mechanism to ensure organizational impact in successful health literacy initiatives [6].

### 4.3. Idea Generation, Co-Designing, and Testing the Intervention

Using co-design throughout the intervention development process is a central feature of the Ophelia approach as a means to increase applicability and penetration. However, in the original protocol [7] and Ophelia manual [26] the participatory elements were not ascribed to any particular methodology. Participatory health research may involve different participatory methodologies within the spectra of action research [44,45]. Many of these are structured similarly to the iterative co-design process of the Ophelia approach [44]. Our results indicate the usefulness of the participatory methodology in integrating different perspectives and creating ownership [6]. Similar results were achieved in other Ophelia projects [11].

For co-design processes, a sample size of 6–12 participants for each activity is recommended [44], which we adhered to in most workshops. However, as we recruited participants for the workshops based on convenience and interest, the sample may not have been representative of the actual user population in the unit. This may have affected the resultant interventions.

Leask et al. (2019) provided a set of recommendations regarding co-creation in public health interventions, including evaluation of such interventions [44]. They recommended evaluating the co-creation process both in terms of validity and co-creator satisfaction along with evaluating the effectiveness of the intervention itself. In this study, validity was evaluated continuously through the iterative process and feedback loops during the intervention development and co-creator satisfaction and ownership was thoroughly evaluated in the organizational impact analysis. From our test cycles, we know that the intervention to a large extent met our predefined objectives. In our organizational impact analysis, we also observed some promising indications about future maintenance, development, and dissemination of the initiatives. The literature on health literacy in relation to CR is scarce but suggests a possible association between health literacy and participation [46], as well as adherence [24] and learning outcomes [25]. Thus, our results call for a future larger scale effectiveness trial of the intervention. 

### 4.4. Strengths and Limitations

The combined methodologies used to describe the application of the Ophelia approach and the evaluation of the organizational impact is a major strength of this study and provides new knowledge on the potential of the Ophelia approach. 

Another strength is the study setup involving staff and user representatives in all steps of the process including the overall coordination. This approach is likely to have increased local ownership and enhance the potential for long-term sustainability [47]. 

The structured intervention development was based on validated health literacy measures and methodologies described in the Ophelia manual [26]. This ensured that the process is reproducible. Our user and organizational needs assessment was more comprehensive than the original Ophelia recommendations. This proved particularly valuable in a setting that was not familiar with the concept of health literacy. 

The intervention was tested with very small sample sizes, providing insight into feasibility and possible outcomes, but we have not been able to assess the effect on rehabilitation outcomes or long-term sustainability. A large-scale trial would provide more information regarding intervention effect and sustainability.

Each part of the intervention development process also had limitations, which will not be discussed in detail here. Limitations of the needs assessments are reported elsewhere [27,34]. The user health literacy assessment produced seven clusters of which only three were described in the vignettes. This may have affected the discussions in the co-design workshops and limited the number of identified intervention ideas. 

In terms of data collection for the organizational impact analysis, we chose to conduct individual interviews for the leaders and user representatives as they were not part of day-to-day practice in the unit. We used these two different approaches to acknowledge the participants’ different qualifications in sharing their experiences, as staff members contrary to leader and user representatives experienced the entire development, testing, and implementation process through their daily practice of the CR Team. However, the staff group still held very different positions within the team. The moderator (C.B.S.) sought to minimize the inherent risk of one participant voice dominating the others [48] by continuously encouraging and directly appealing to all participants to contribute [48]. 

Our closed coding procedure in the data analysis had clear advantages in relation to evaluating the application of the Ophelia approach, but at the same time such a strategy precludes unexpected findings outside the predefined areas of interest [35]. Other evaluation frameworks might have led to different results. In future scaled-up intervention evaluations it would also be relevant to include clinical measures such as rehabilitation outcomes and in a longer perspective cardiovascular risk and disease outcomes.

In summary, our findings support the use of the Ophelia approach in developing initiatives to improve health literacy responsiveness. Using the approach not only produced feasible interventions, but our results also indicate that the organizational values, strategies, and practices were affected by the process in a way that is likely to support long-term sustainability. Furthermore, all CR staffs and relevant leaders were involved, which is likely to have further increased local ownership. Thus, we recommend the use of the Ophelia approach in future intervention studies aimed at improving health literacy responsiveness, and encourage research evaluating its application in larger settings.

## 5. Conclusions

In a municipal CR unit the Heart Skills Study improved organizational health literacy responsiveness by developing a feasible intervention using an extended version of the Ophelia approach. Applying the Ophelia approach also had a substantial organizational impact in the unit, improving its health literacy responsiveness further by developing and integrating a new and more holistic understanding of user needs and vulnerabilities based on health literacy and by creating a familiarity with the use of participatory methodologies and improving focus on differentiated healthcare practices. 

Our findings can be used to inform the development and evaluation of sustainable co-designed organizational health literacy responsiveness initiatives in other settings. 

## Figures and Tables

**Figure 1 ijerph-17-01015-f001:**
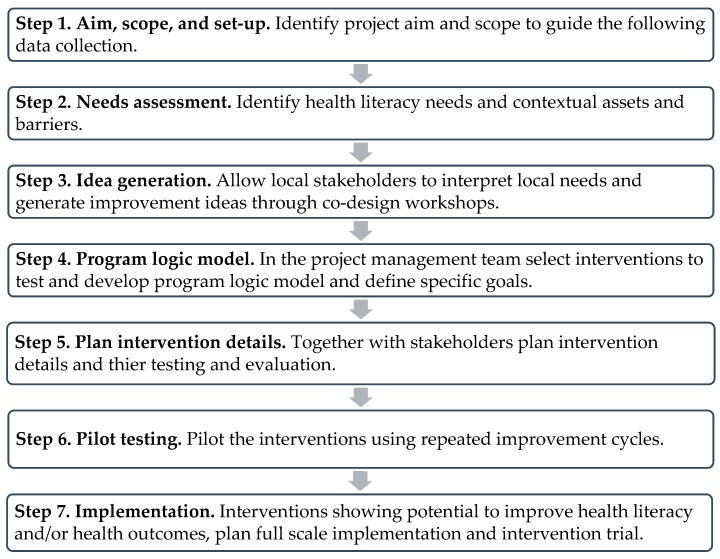
The Ophelia approach in seven steps (adapted from Batterham et al. [7] *).* The steps were adapted in accordance with the steps described in the Ophelia manual [26], e.g., adding step 1 to the original methodology.

**Figure 2 ijerph-17-01015-f002:**
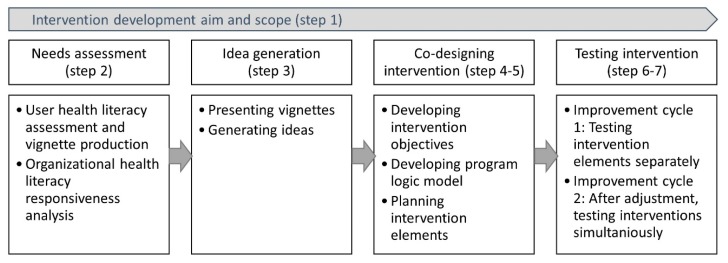
Application of the extended Ophelia approach to develop a health literacy responsiveness intervention in Randers Municipal Rehabilitation Unit 2017–2019.

**Table 1 ijerph-17-01015-t001:** Eight Ophelia principles to guide the development and implementation of interventions (adapted from Beauchamp et al. [11] *).

Principle	Explanation
Outcomes focused	Improve health and reduce health inequities, e.g., by meeting project aims and intervention objectives, and implementing logic models.
Equity driven	All activities at all stages prioritize disadvantaged groups and those experiencing inequity in access and outcomes, e.g., by identifying and acting upon the needs of disadvantaged groups.
Co-design approach	In all activities at all stages, relevant stakeholders engage collaboratively to design solutions.
Needs-diagnostic approach	Participatory assessment of local needs using local data, e.g. using multidimensional health literacy tools.
Driven by local wisdom	Intervention development and implementation is grounded in local experience and expertise.
Responsiveness	Organizational response to health literacy diversity and other unique needs in the target population takes account of individuals, contexts, cultures and time
Systematically applied	A multilevel approach in which resources, interventions, research and policy are organized to optimize health literacy, e.g., by improving client’s skills, enabling clinicians, changing organizational processes or engaging with external agencies.
Sustainable **	Optimal health literacy practice becomes normal practice and policy, e.g., when small interventions at one level build up over time to achieve organizational priorities and objectives.

* Details and examples were added to the original version. ** In the original publication of the principles ‘sustainable’ was listed before ‘systematically applied’. We changed the order to support the logic of our reporting, ending with long-term sustainability.

**Table 2 ijerph-17-01015-t002:** Socio-demographic, health, and health literacy characteristics by cluster (*n* = 162) in the Heart Skills Study survey (2017).

Cluster	1 & 2 *	3	4	5	6	7	All
n in Cluster	29	64	10	18	26	15	162
% of total population	17.90	39.51	6.17	11.11	16.05	9.26	100.00
**Socio-demographic characteristics**							
Mean age (years)	67.86	65.95	71.87	68.88	66.26	63.53	66.78
(SD)	(10.03)	(11.53)	(10.66)	(10.72)	(11.03)	(13.89)	(0.91)
Male gender	22	46	10	N/A	17	8	N/A
(%)	(75.86)	71.88)	(100.00)	N/A	(65.38)	(53.33)	N/A
Living alone	N/A	17	N/A	5	5	7	42
(%)	N/A	(27.87)	N/A	(27.78)	(20.00)	(46.67)	(26.58)
≤11 years of schooling	5	10	N/A	N/A	5	5	30
(%)	(17.24)	(17.86)	N/A	N/A	(20.00)	(38.46)	(20.83)
**Health characteristics**							
Not participating in rehabilitation	N/A	10	0	5	N/A	5	24
(%)	N/A	(15.87)	(0.00)	(27.78)	N/A	(33.33)	(14.91)
Smoker	6	19	N/A	N/A	6	7	44
(%)	(20.69)	(30.65)	N/A	N/A	(23.08)	(46.67)	(27.50)
Mean HRQoL (physical component summary)	44.00	40.40	41.80	41.95	37.93	36.59	40.49
(SD)	(10.38)	(11.03)	(11.50)	(9.52)	(10.45)	(9.99)	(0.86)
Mean HRQoL mental component summary)	51.81	48.82	42.27	48.91	38.87	43.50	46.76
(SD)	(9.10)	(9.92)	(13.30)	(13.00)	(9.39)	(10.26)	(0.89)
**Health literacy (mean Scale Scores)**							
1. Healthcare provider support	3.73	2.92	3.33	3.19	2.52	2.67	3.03
(SD)	(0.49)	(0.42)	(0.35)	(0.39)	(0.39)	(0.43)	(0.57)
2. Having sufficient information	3.61	3.12	3.20	2.88	2.56	2.42	3.03
(SD)	(0.57)	(0.30)	(0.28)	(0.27)	(0.34)	(0.36)	(0.52)
3. Actively managing health	3.52	3.01	3.02	2.39	2.87	2.25	2.94
(SD)	(0.38)	(0.25)	(0.15)	(0.33)	(0.25)	(0.38)	(0.47)
4. Social support for health	3.66	3.10	3.52	2.99	2.75	2.89	3.14
(SD)	(0.52)	(0.35)	(0.41)	(0.46)	(0.57)	(0.43)	(0.53)
5. Appraisal of health information	3.28	2.76	3.16	2.12	2.57	2.05	2.71
(SD)	(0.42)	(0.31)	(0.30)	(0.45)	(0.34)	(0.32)	(0.52)
6. Active engagement	4.38	3.88	3.32	3.98	2.96	3.16	3.73
(SD)	(0.54)	(0.35)	(0.45)	(0.38)	(0.46)	(0.61)	(0.65)
7. Navigating the health system	3.99	3.64	2.88	3.58	2.64	2.41	3.37
(SD)	(0.60)	(0.46)	(0.41)	(0.43)	(0.50)	(0.57)	(0.73)
8. Finding health information	4.14	3.88	3.06	3.60	3.06	2.34	3.57
(SD)	(0.51)	(0.30)	(0.34)	(0.50)	(0.47)	(0.45)	(0.68)
9. Understanding health information	4.05	3.90	3.00	3.86	3.25	2.60	3.64
(SD)	(0.53)	(0.30)	(0.31)	(0.47)	(0.42)	(0.71)	(0.63)

SD, standard deviation; HLQ, Health Literacy Questionnaire; HRQoL, health-related quality of life, measured using the Short Form Health Survey 12 (SF-12) and its component scores [36]; N/A, not available due to data protection regulations; * clusters 1 and 2 were merged post-analysis due to data protection considerations.

**Table 3 ijerph-17-01015-t003:** Intervention objectives related to the initial phases of the cardiac rehabilitation (CR) program and results of the second quality improvement cycle.

Intervention Objective	Test Result
In the test period the number of referred people declining CR are reduced by 25% compared to survey data from 2017	Of 33 participants in start-up sessions in the test period 2 (6.1%) declined further rehabilitation. In the 2017 survey 25/174 (14.4%) reported non-participation–it is not known how many of these attended start-up sessions.
Before the test period, resources and support to encourage relatives and friends to participate in the rehabilitation program is developed	A written information leaflet was produced based on consultations with users and their relatives. Also, verbal invitation of relatives and friends has been introduced in the initial telephonic contact with people referred to the unit.
In the test period 50% of people attending their CR start-up session bring a relative, friend or lay counsellor	Out of 33 referrals, 18 (54.5%) brought a relative or friend to the start-up session.
Before the test period a resource to support the problem-based needs assessment and planning session is developed	To identify vulnerable CR attendants, the “Conversational Health Literacy Assessment Tool” [37,38] was introduced and implemented. A consultation guide was developed to support the problem-based needs assessment and planning session.
In the test period vulnerable users are successfully identified and offered a problem-based needs assessment and planning session	Out of 31 rehabilitation starters 4 (12.9%) were identified as vulnerable and all were offered the problem-based needs assessment and planning session

CR, cardiac rehabilitation.

**Table 4 ijerph-17-01015-t004:** Intervention packages developed in the Heart Skills Study in Randers Municipal Rehabilitation Unit (2017–2019).

Package	Aim	Content of Package
1	Improve the social support of all people referred to CR in the unit	1. Handing out written information at the regional hospital aimed at supporting relatives or friends. 2. Verbally invite users to bring a relative or friend to CR start-up sessions. 3. In collaboration with a local lay counselor association, offer a voluntary ‘substitute relative’ when a referred person has no relevant person to bring.
2	Identify and respond to the needs of vulnerable people referred to the rehabilitation program	1. Identify vulnerability based on a negative assessment of health literacy using the “Conversational Health Literacy Assessment Tool” [38], identification of mental challenges using the “Hospital Anxiety and Depression Scale” [39] or ‘at risk of non-adherence’ to the rehabilitation program. 2. Offer an extra problem-based needs identification and program planning session to the vulnerable group. This involves discussion of their general situation and challenges, leading to an individualized care plan.

CR cardiac rehabilitation.

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
