# Peer review of "Improving Organizational Health Literacy Responsiveness in Cardiac Rehabilitation Using A Co-Design Methodology: Results from The Heart Skills Study"

_ijerph, 2020, doi:10.3390/ijerph17031015_

Round 1
Reviewer 1 Report
Thank you for giving me the opportunity to review this paper.
A focus on the development of organisational health literacy is an important contribution to the literature and the application to cardiac rehab patients is both important and useful.
My over-riding concern is that there is perhaps too much overlap between this paper and previously published data from this work. The data presented in table 2 reports on the same data set included in the article: Aaby et al (2020) . Health Literacy among People in Cardiac Rehabilitation: Associations with Participation and Health-Related Quality of Life in the Heart Skills Study in Denmark.IJERPH https://www.mdpi.com/1660-4601/17/2/443 While it is not reported according to cluster in the previous paper a clear justification needs to be made as to why this data is needed here again and in this format.
The adaptation of the Ophelia approach is to include the HL responsiveness analysis is potentially important. Some additional context as to why this particular approach is added would be helpful.
Some more detail about sampling throughout is needed. Did the 222 participants in the HL survey represent the total population group or sample selected from it? How are staff reps and leaders defined and identified. Again, how does the number selected represent the total population groups?
The evaluation of the Ophelia approach is the most useful part of this paper and is the most unique aspect of it. However, a more detailed explanation and justification of the evaluation approach would strengthen this. It is implied that what is being evaluated is the application of Ophelia principles. Why is the this the focus of the evaluation as opposed to the use of impact and outcome indicators?
Why were some professionals interviewed and others involved in focus groups?
How were the deductive codes for analysis of the interviews selected?
Unfortunately I cannot open the supplementary file with the interview questions on. Please can this be reattached.
Author Response
Response to reviewer comments
Dear reviewers,
Thank you for your general appreciation and the many appropriate inputs for the study “Improving organizational health literacy responsiveness in cardiac rehabilitation using a co-design methodology: results from the Heart Skills Study”.
In the following, we provide a point-to-point reply to all comments. Please, see also the track changes in the revised manuscript.
We hope we have provided a satisfactory response, and will be glad to address any further comments that you may have.
Kind regards.
Reviewer 1:
|
Reviewers comments |
Our response |
|
My over-riding concern is that there is perhaps too much overlap between this paper and previously published data from this work. The data presented in table 2 reports on the same data set included in the article: Aaby et al (2020). Health Literacy among People in Cardiac Rehabilitation: Associations with Participation and Health-Related Quality of Life in the Heart Skills Study in Denmark. IJERPH https://www.mdpi.com/1660-4601/17/2/443 While it is not reported according to cluster in the previous paper a clear justification needs to be made as to why this data is needed here again and in this format. |
Thank you for this comment. We fully understand the concern, but we also believe that the fact that the two papers build on slightly different samples should be considered. In the methods section, we have tried to clarify the reasons for including some of the same data (cf. ll. 116-118 and ll. 127-130 of the revised manuscript). It is true that the survey among 222 people referred to CR in 2017 provides the data foundation of both analyses. However, in the association study only a subpopulation (n=150) referred between March and December 2017 is included. In the cluster analysis all respondents providing enough HLQ information (N = 162) was included. The cluster analysis is merely a descriptive analysis comparable to table 2 of the IJERPH https://www.mdpi.com/1660-4601/17/2/443 though as mentioned based on a different subpopulation. It does however provide important information on the health literacy diversity across a population referred to the CR program in Randers Municipality based on which the co-design process was carried out. We therefore believe removing the cluster analysis (table 2) from the manuscript would reduce the understanding of how to conduct a co-design process building on local data. |
|
The adaptation of the Ophelia approach is to include the HL responsiveness analysis is potentially important. Some additional context as to why this particular approach is added would be helpful. |
We agree that our argumentation could have been clearer and we have added further information in the discussion section (cf. ll. 483-487 in the revised manuscript). |
|
Some more detail about sampling throughout is needed. Did the 222 participants in the HL survey represent the total population group or sample selected from it? How are staff reps and leaders defined and identified. Again, how does the number selected represent the total population groups? |
We agree that more detail could be added on sampling for the different activities in the intervention development process. We have done so in the following way: 1) Regarding the 222 participants in the survey, they represented all individuals referred to the CR unit in 2017. This is already stated in the methods section, where we have now made a few corrections to make it clearer (cf. ll. 116-117 and ll. 229-230) 2) Regarding staff and managers involved in the co-design workshops, we have added details on their selection in the methods section (cf. ll. 156-157, ll. 171-172, and ll. 179-181) and the discussions section (cf. ll. 500-502) in the revised manuscript. 3) Regarding recruitment for the qualitative interviews, we have added details in the methods section (cf. ll. 188-191). |
|
The evaluation of the Ophelia approach is the most useful part of this paper and is the most unique aspect of it. However, a more detailed explanation and justification of the evaluation approach would strengthen this. It is implied that what is being evaluated is the application of Ophelia principles. Why is the this the focus of the evaluation as opposed to the use of impact and outcome indicators? |
We agree that the arguments may be a little vague. We have extended the argumentation in the methods section (cf. ll. 186-187). We also acknowledge your point in the limitations section of the discussion (cf. ll. 546-551). The impact and outcome indicators of the process would indeed be relevant to study, however this was not the scope of this work. When scaling-up interventions in the unit it would be relevant to define outcomes with the local participants such as rehabilitation outcomes and (if the timeframe allows) CVD outcomes. |
|
Why were some professionals interviewed and others involved in focus groups? |
We argue in the methods section (cf. ll. 187-189) and in the discussions section (cf. ll. 537-542), that we have used the two different data collection methodologies to acknowledge the participants different qualifications in sharing their experiences as participants and observers of the application of the Ophelia approach. The argument has been extended in the revised manuscript in the discussions section (cf. ll. 537-542). |
|
How were the deductive codes for analysis of the interviews selected? |
The codes were the eight Ophelia principles – so they were not selected as such, but predefined by the developers of the Ophelia approach. We have made this clearer in the methods section (cf. ll. 197-200). |
|
Unfortunately I cannot open the supplementary file with the interview questions on. Please can this be reattached. |
They are all reattached as PDF’s in a zipfile as required by IJERPH. Please let us know if you experience any further trouble with the files. |
Reviewer 2 Report
Overall, a well planned and executed study on “Improving organizational health literacy rehabilitation using a co-design methodology”. The study has clearly been described and is generally free from any major or remarkable errors. However, there were noticeable typos and grammatical errors in some sections of the manuscript as stated below.
Introduction – Good literature review, well described, otherwise unremarkable.
Material & Methods – Extensive description of the methodology.
Results – Clearly stated.
Correct grammatical errors on:
Page 6, line 219-220 – one of the Cluster(s) (add ‘s’).
Page 8, line 243 – An example(s), (delete ‘s’)
Page 8, line 246 – Another example, (add ,)
Page 9, line 285, Table 4 – declining CR declines (preferably, use ‘reduces’ instead)
Discussion – Page 13, line 460 – replace ‘faces’ with ‘faced’
Conclusion – Well described.
References – Unremarkable.
Author Response
Response to reviewer comments
Dear reviewers,
Thank you for your general appreciation and the many appropriate inputs for the study “Improving organizational health literacy responsiveness in cardiac rehabilitation using a co-design methodology: results from the Heart Skills Study”.
In the following, we provide a point-to-point reply to all comments. Please, see also the track changes in the revised manuscript.
We hope we have provided a satisfactory response, and will be glad to address any further comments that you may have.
Kind regards.
Reviewer 2:
|
Reviewers comments |
Our response |
|
Correct grammatical errors on: Page 6, line 219-220 – one of the Cluster(s) (add ‘s’). Page 8, line 243 – An example(s), (delete ‘s’) Page 8, line 246 – Another example, (add ,) Page 9, line 285, Table 4 – declining CR declines (preferably, use ‘reduces’ instead) Discussion – Page 13, line 460 – replace ‘faces’ with ‘faced’ |
Thank you for the grammatical corrections. The errors have all been corrected. A new version of table 4 is attached as required. |
Other corrections:
We have added further details on the development of vignettes as we felt that the original manuscript did not fully elucidate this process (cf. methodology section ll. 133-140).